# Failure Mechanism of Gun Barrel Caused by Peeling of Cr Layer and Softening of Bore Matrix

**Zi-meng Wang [1], Chun-dong Hu [1],\*, Yang-xin Wang [1], Heng-chang Lu [1],\*, Jun-song Li [2] and Han Dong [1]**

[1] School of Materials Science and Engineering, Shanghai University, Shanghai 200444, China; wangzimeng@shu.edu.cn (Z.-m.W.); flypenguin@shu.edu.cn (Y.-x.W.); Donghan@shu.edu.cn (H.D.)

[2] No. 208 Institute of China Ordnance Industry, Beijing 102202, China; lijunsong208@shou.com

\* Correspondence: huchundong99@163.com or hucd99@shu.edu.cn (C.-d.H.); luhengchang@shu.edu.cn (H.-c.L.); Tel.: +86-131-6209-5128 (C.-d.H.)

**Abstract:** Research on failure mechanism is essential for the prolonging of gun barrel lifetime. To explore the gun barrel failure mechanism, the damage characteristics of a machine gun barrel were characterized. The results show that the failure of the gun barrel is correlated with the peeling of the Cr layer on the bore surface and the softening of the bore matrix. The damage of the Cr layer varies along the axial direction. From the gun tail to the muzzle, the damage mode of the Cr layer changes from peeling to wearing. The damage rate in both the tail and the muzzle is higher than that in the middle of the barrel. The matrix close to the bore surface is softened due to the high temperatures caused by stress–relief–annealing and shooting. The gun tail suffers from higher temperatures, thus being softened more seriously than the other parts. The softened matrix results in an increase in the tendency of plastic deformation of the bore surface and an increment in the bore diameter, which leads to a decrease in the firing accuracy.

**Keywords:** gun barrel; failure mechanism; crack propagation; bore matrix softening; 3D intelligent hyperfield microscopy

## 1. Introduction

It has been widely acknowledged that the failure of a gun barrel is correlated to thermal, chemical, or mechanical actions. The enlargement of bore diameter could lead to a reduction in both muzzle velocity, range and a decrement in firing accuracy [1–5].

The research on the failure mechanism of the gun barrel has been receiving increasing interest. Cote et al. [6] proposed a model describing the mechanism of gas–solid erosion, which indicates that the erosion is mainly caused by the reaction between gunpowder exhaust and matrix under high temperature and pressure conditions. Lawton et al. [7] correlated the wear of bore surface with initial temperature, maximum surface temperature, and gunpowder composition. Their study indicates the importance of heat and gunpowder composition on erosion. Sopok et al. [8–10] developed a thermo–chemical–mechanical erosion model, defining different erosion areas, and annotating different erosion mechanisms. Underwood et al. [11–14] analyzed gun barrel damage through a thermodynamic model and a fracture model. Qiao et al. [15] investigated the bore damages in a failed large-caliber machine gun barrel and found that the bore surface in the tail region was largely damaged by erosion and that the damage in the four and five cones is dominant for the failure of the gun barrel.

The bore damage caused by erosion and/or wear during shooting is a vital factor influencing the barrel life. According to the forementioned mechanisms of erosion and wear, the damage degree and mode of the Cr layer on the bore surface vary along the axial direction. In our previous studies [16,17], the Cr layer on the groove lines in the tail region was observed to be damaged prior to that on the land lines. The damage of the Cr layer

aggravates the damage of the matrix and increases the tendency toward enlarging the bore diameter.

In this paper, 3D intelligent hyperfield microscopy (3DIHM) was for the first time introduced into the study of gun barrel damage. With 3DIHM, the damage characteristics of a machine gun barrel were characterized and the corresponding failure mechanisms discussed.

## 2. Materials and Methods

The material used in this study was a Cr–Ni–Mo–V steel with chemical compositions (in wt.%): $0.31 \pm 0.008$ C, $0.25 \pm 0.01$ Si, $0.24 \pm 0.01$ Mn, $0.009 \pm 0.001$ P, $0.004 \pm 0.001$ S, $1.92 \pm 0.01$ Cr, $1.42 \pm 0.009$ Mo, and $0.31 \pm 0.01$ V. Each firing cycle was carried out at three temperatures: room temperature, high temperature ($50 \pm 2$ °C) and low temperature ($-49 \pm 2$ °C). The bore diameter was measured with a ruler gauge, for which the measurement method can be found in the literature [15]. When the projectile volume was 0, 7, 24, ..., or 100% of the full life, they were marked as 0% life, 7% life, 24% life, ..., or 100% life. The criteria for the test of gun barrel life were as follows: (i) the reduction rate of initial muzzle velocity was greater than or equal to 15%; (ii) the number of elliptical bullet holes (the ratio of major axis to minor axis is greater than 1.25) which exceeded 50% of the projectile number; (iii) the average dispersion density of three consecutive targets was $R50 \geq 30$ cm; and (iv) visible cracks appeared. Provided that one of the criteria was reached, the gun barrel was declared to be failed.

The barrel used in this study was a large-caliber machine gun barrel. To explore the damage characteristics of the gun barrel, seven samples were taken with the positions being 113–123 mm, 133–233 mm, 243–253 mm, 343–353 mm, 543–553 mm, and 743–753 mm, respectively (the tail as the starting point). The samples were named MG1, MG2, MG3, MG4, MG5, MG6, and MG7, correspondingly (see Figure 1). It should be noted that only MG2 was a longitudinal sample and the rest were ring samples.

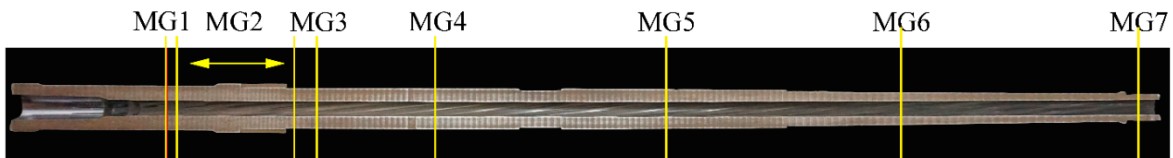

**Figure 1.** Sample positions in a large-caliber machine gun barrel.

All samples were first cleaned by ultrasonic wave in an acetone solution, and then the ring samples were ground and polished after being inlaid in epoxy resins. The cross-sectional morphologies of the samples were observed with Smart Zoom 5 3DIHM manufactured by Carl Zeiss company (Oberkochen, Germany). A Nova NanoSEM 450 scanning electron microscope (SEM) produced by FEI company (Hillsboro, OR, USA) was used for the characterization of the microstructures of the samples. All samples for the SEM test were corroded with 4% nitric acid alcohol solution. Both alcohol and nitric acid were made by China National Pharmaceutical Group Corporation (Beijing, China). Then 4% nitric acid alcohol solution was prepared in proportion. A Wilson hardness tester was used for the measurement of radial Vickers hardness, with a load of 1000 g. The hardness of MG1, MG5 and MG7 in eight directions (not including the Cr layer) was tested.

## 3. Results

### 3.1. Bore Diameter Variation along Axial Direction

The variation of bore diameter of the gun barrel, spanning 0–100% life, is shown in Figure 2. For the convenience of depiction, the whole barrel was divided into three parts: the tail, middle, and muzzle. According to this division, MG1 and MG2 are in the tail, MG3, MG4, MG5, and MG6 are in the middle, and MG7 is in the muzzle. From the tail to the muzzle, the bore diameter decreases first and then increases. With the increasing gun

life, the increment in the bore diameter in both the tail and muzzle regions is larger than that in the middle part.

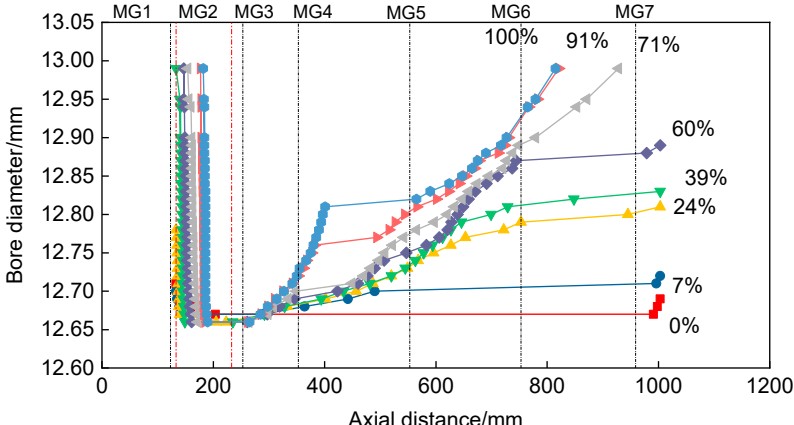

**Figure 2.** Variation of bore diameter of a gun barrel during the 0–100% life span.

### 3.2. Bore Surface Damage

The cross-sectional morphologies of MG1, MG3, MG5, and MG7 taken by 3DIHM are shown in Figure 3. The red circle in the pictures represents the initial bore diameter, which is 12.7 mm. Before shooting, the whole bore surface is covered with a Cr layer. Figure 3a shows that MG1 (in the tail) presents a state of serious damage, where cracks extend deeply into the matrix from the bore edge with the long cracks being uniformly distributed. No Cr layer remains, and the bore diameter is obviously expanded. As shown in Figure 3b, the damage degree of MG3 (in the middle) is significantly reduced. The Cr layer on the grooves falls largely off. The cracks in the exposed matrix are short and dense. The maximum thickness of the Cr layer on the lands of MG3 is about 190 ± 4 µm. The damage degree of MG5 is further reduced (see Figure 3c). Only a small part of the Cr layer on the grooves falls off and only a small quantity of short cracks is observed. The maximum thickness of the Cr layer on the lands of MG5 is about 100 ± 4 µm. In MG7 (Figure 3d), the lands are largely worn off, leaving the remainder being aligned to the Cr layer on the grooves. In MG7, the Cr layer is only observed on the grooves and almost no cracks are present.

To characterize the degree of bore surface damage, a rifling retention $X$ is defined as

$$S_{\min} = \frac{(d_{\min} - 12.7)}{12.7} \times 100\% \tag{1}$$

$$S_{\max} = \frac{(d_{\max} - 12.7)}{12.7} \times 100\% \tag{2}$$

$$X = \frac{S_{\max}}{S_{\min}} \tag{3}$$

where $S$ is the expansion rate of bore diameter and $d$ is the bore diameter. The subscripts "min" and "max" refer to the minimum and maximum states, respectively. The value of $X$ reflects the degree of rifling retention, where a larger value indicates a better degree of rifling retention.

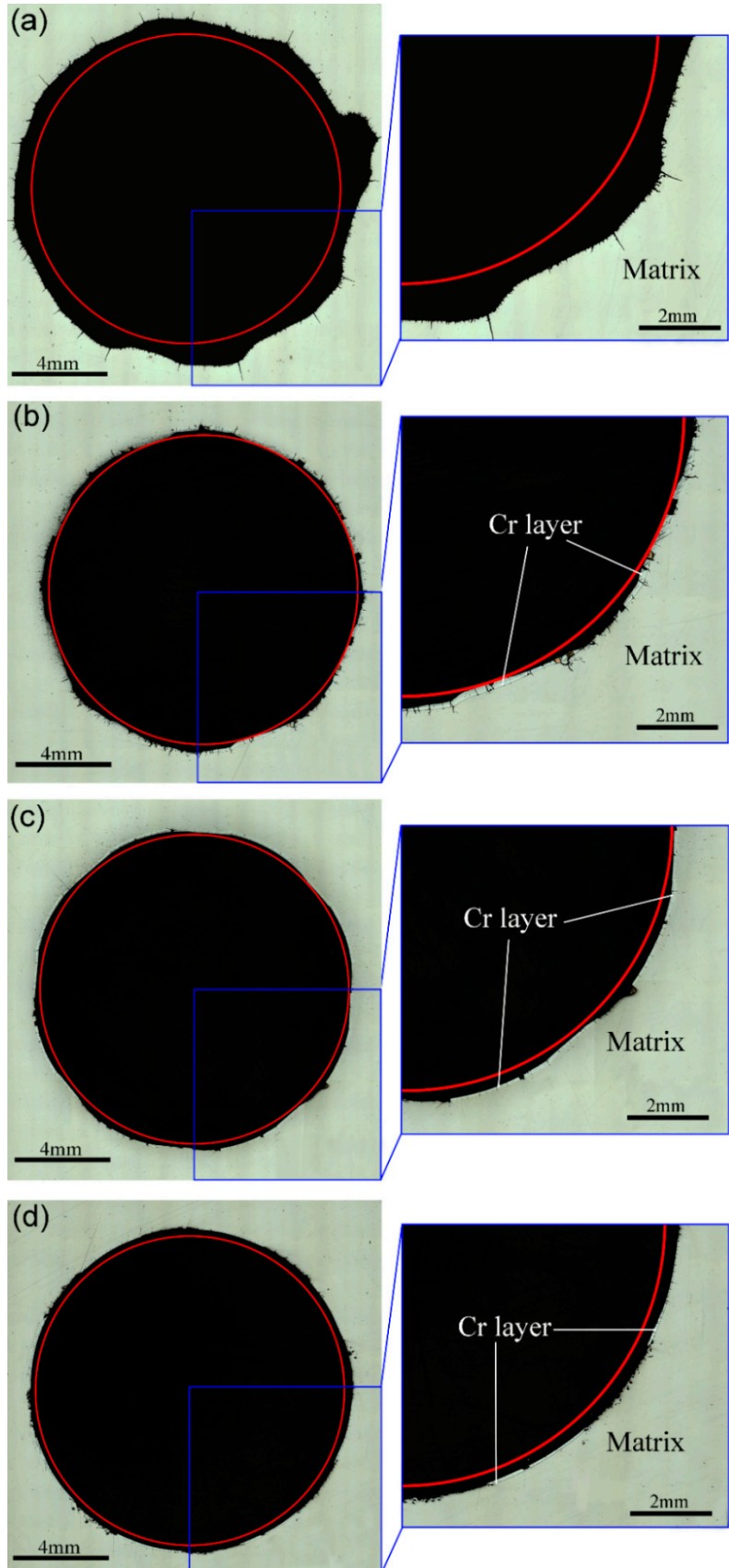

**Figure 3.** Cross-sectional morphologies of the samples taken by 3D intelligent hyperfield microscopy (3DIHM): (**a**) MG1; (**b**) MG3; (**c**) MG5; and (**d**) MG7.

The calculated values of $X$ for all ring samples are shown in Table 1. The results show that the values of $X$ in both the tail and muzzle are small, indicating a comparatively bad rifling retention in these two parts. The values of $X$ in the middle of the gun are large, implying that the rifling retention in this part is good. Both values of $S_{min}$ and $S_{max}$ reflect the damage degree of the bore surface, where the larger the values are, the more serious the damage is. The data in Table 1 show that the values of $S_{min}$ and $S_{max}$ in both the tail and muzzle are larger than those in the middle, indicating that the damage degree in both the tail and muzzle is obviously greater than that in the middle. This is consistent with the variation of bore diameter.

**Table 1.** Experimental values of $d_{min}$, $S_{min}$, $d_{max}$, $S_{max}$, and $X$.

| Sample | $d_{min}$/mm | $S_{min}$/% | $d_{max}$/mm | $S_{max}$/% | $X$ |
|--------|--------------|-------------|--------------|-------------|-----|
| MG1 | 13.76 | 8.3 | 15.31 | 20.5 | 2.4 |
| MG3 | 12.71 | 0.1 | 13.33 | 4.9 | 49 |
| MG4 | 12.73 | 0.2 | 13.43 | 5.7 | 28.5 |
| MG5 | 12.86 | 1.2 | 13.42 | 5.6 | 4.6 |
| MG6 | 12.94 | 1.8 | 13.25 | 4.3 | 2.3 |
| MG7 | 13.19 | 3.8 | 13.31 | 4.8 | 1.2 |

The gun tail shows the highest damage rate, where the bore diameter increases sharply with only a small number of projectiles. The morphology of MG1 characterized by SEM is shown in Figure 4. The cracks are evenly spaced on the bore surface and are rich in elements S, Cu and Pb. The element Cu may come from the Cu layer of bullet shells.

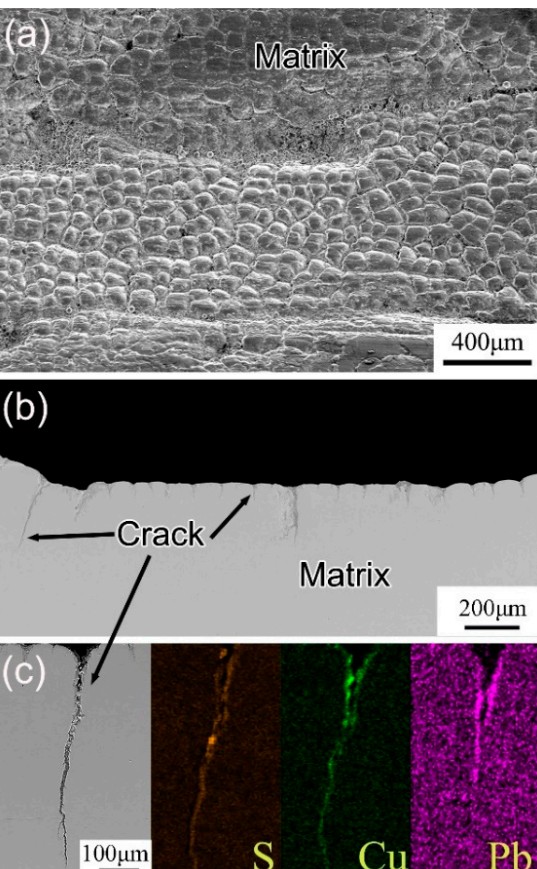

**Figure 4.** SEM morphologies of MG1: (**a**) bore surface; (**b**) cracks in the cross-section; and (**c**) the EDS energy spectrum of a crack.

The morphologies of MG2 characterized by 3DIHM are shown in Figure 5. The results show that, from the left to the right (namely from the part near the tail to the part close to MG3), there is more and more remaining Cr layer. The surface of the matrix near the gun tail is full of cracks (see Figure 5a).

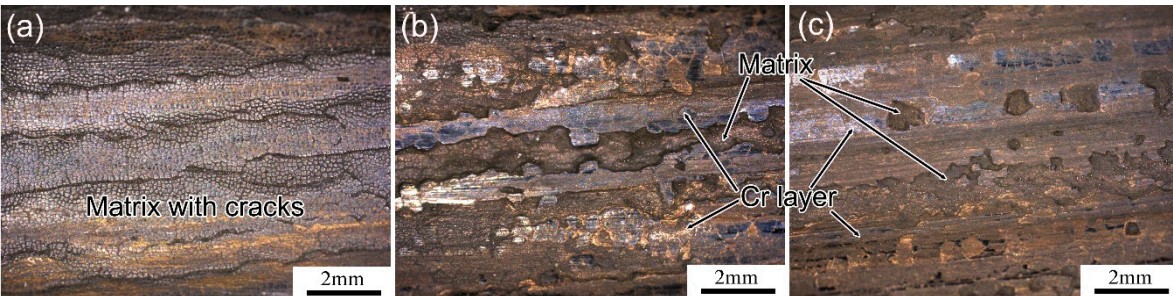

**Figure 5.** Longitudinal profile morphologies of MG2: (**a**) the part close to the tail; (**b**) transitional part; and (**c**) the part close to MG3.

The SEM morphologies of MG3 are shown in Figure 6. The MG3 is at a position where the bore diameter is smallest and the rifling retention is best. Only a small amount of Cr layer was peeled off. The EDS spectrum shows that the cracks are rich in S, Pb and Cu elements.

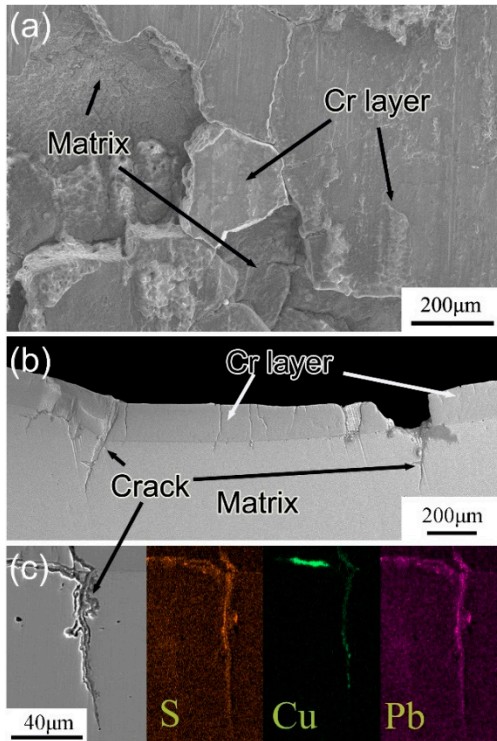

**Figure 6.** SEM morphologies of MG3: (**a**) bore surface; (**b**) cracks in cross-section; and (**c**) EDS energy spectrum of a crack.

The cross-sectional morphology of MG7 is shown in Figure 7. The Cr layer on the leading side of the land line is worn off, resulting in an exposure of the matrix which is aligned to the Cr layer on the groove. The exposed matrix also suffers from an obvious wear. The damage in this area is primarily caused by the wearing induced by bullets, while the peeling of the Cr layer is relatively light.

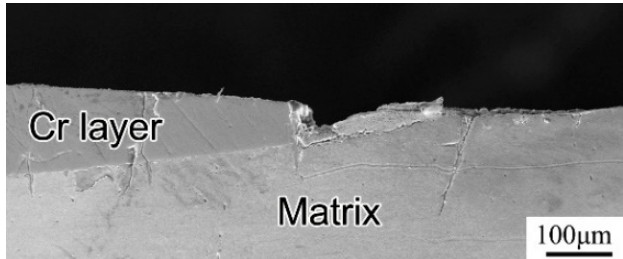

**Figure 7.** Cross-sectional SEM morphology of MG7.

The experimental values of the depth and spacing of the cracks on the bore surface of MG1 and MG5 are shown in Table 2. The Cr layer in MG1 is completely peeled off. The average crack depth of MG1 without the Cr layer is 37 μm and the average crack spacing is 40 μm. In MG5, the crack depth of the matrix with Cr layer is shallow with an average value of 32 μm, while the crack depth of the matrix without the Cr layer is comparatively larger (50 μm). This indicates that the Cr layer can inhibit the propagation of cracks. The crack spacing of the matrix containing the Cr layer is smaller than that of the matrix without the Cr layer. In MG7, the damage is mainly caused by wear, and the matrix has almost no cracks.

**Table 2.** Experimental values of crack depth and spacing in a cross-sectional matrix.

| Region | With or without Cr Layer | Average Crack Depth/μm | Average Crack Spacing/μm | Ratio of Average Spacing to Depth |
|---|---|---|---|---|
| MG1 | With | - | - | - |
|  | Without | 37 | 40 | 1.1 |
| MG5 | With | 32 | 87 | 2.7 |
|  | Without | 50 | 92 | 1.8 |

### 3.3. Steel Matrix Hardness

The hardness of MG1, MG5, and MG7 is demonstrated in Figure 8. From the perspective of radial hardness distribution, the hardness of the matrix near the bore surface is lower than that near the edge. From the tail to the muzzle, the hardness shows an increasing trend.

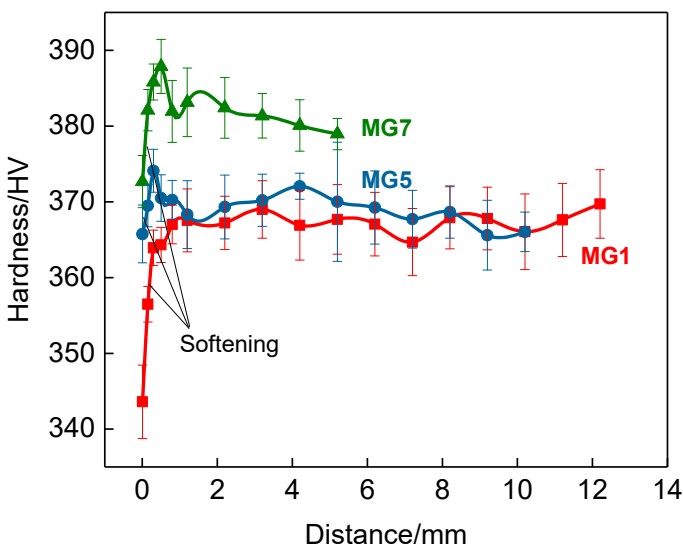

**Figure 8.** Hardness of MG1, MG5, and MG7.

## 4. Discussion

### 4.1. Damage Characteristics of Bore Surface

The damage mode of the gun barrel varies along the axial direction. This is because the acting conditions such as temperature, stress, and corrosion on the bore surface are constantly changed along the axial direction.

At the tail of the barrel, the damage mode of the bore surface is as follows: the Cr layer falls off, with the exposed matrix being covered with crack networks. This phenomenon was also observed by Li et al. [18] in a hot-stamping boron steel with Al–Si coating. Cyclic high temperature and pressure caused by shooting leads the Cr layer to compress and yield, and brittle cracks occur during the cooling process [19]. With the cyclic tensile loading, low cycle fatigue failure occurs [20–22].

According to the stress analysis based on the linear elastic theory, when the Cr layer is under a load and the maximum tensile stress is close to the yield strength of the base steel, the Cr layer is easy to crack [23]. The tensile stress $\sigma$ in the Cr layer can be estimated according to the following formula:

$$\sigma = (\frac{E'}{E})P(r_2^2 + r_1^2)/(r_2^2 - r_1^2) \tag{4}$$

where $E'$ and $E$ are the elastic moduli of chromium and barrel steel, respectively, $r_1$ is the bore radius, $r_2$ is the outer radius of the barrel, and $P$ is the gas pressure caused by gunpowder. For a 12.7 mm machine gun: $E'/E \approx 1.5$, $(r_2^2 + r_1^2)/(r_2^2 - r_1^2) \approx 1.27$, $P = 310$ MPa, thus $\sigma \approx 591$ MPa. The strength of gun steel is 198–600 MPa at 500–700 °C. Within this stress range, the Cr layer is most likely to crack.

Under the condition of high pressure, the gas produced by gunpowder promotes the cracks to propagate continuously. The EDS energy spectra of cracks confirmed this phenomenon (see Figures 4 and 6). Such damage prevails in the tail and the middle of the barrel. At the muzzle, the cracks in the Cr layer are difficult to extend to the matrix because of the reduced pressure.

The width, depth, and density of cracks affect the variation of the bore diameter. The ratio of the average crack spacing to crack depth in the tail and middle parts ranges between 1.1 and 2.7. This is similar to that obtained by Underwood J.H. et al. [11], which is 2. During the cooling process after shooting, the residual tensile stress is formed in the coating. Due to the formation of the initial main cracks, the macro stress caused by the residual tensile stress is reduced, and it is in balance with the interfacial shear stress in the $L$-length range. According to slip zone theory, Underwood et al. analyzed the cracking of the Cr layer, and gave a mathematical expression of crack depth $h$ and crack spacing $L$:

$$\frac{L}{h} = \frac{S_{\text{resid}}}{\tau_y} = \frac{S_y}{\tau_y} \tag{5}$$

In the formula, $S_{\text{resid}}$ is the residual tensile stress, $\tau_y$ is the interfacial shear stress, and $S_y$ is the tensile yield strength. For the Cr layer, the general yield strength $S_y$ is twice that of the shear stress $\tau_y$, thus the value of $L/h$ is approximately 2.

In the middle part of the barrel, the Cr layer was partially peeled off, with the bore diameter varying only slightly. Compared with the condition in the tail, both the temperature and pressure in the middle of the barrel were reduced. Thus, the damage of the Cr layer is less and the damage rate slows down. Qiao et al. [15] believed that the middle part of the barrel was an important area for projectile rifling etching and has a significant impact on shooting accuracy.

The damage characteristics in the muzzle part are as follows: the Cr layer on the land lines are worn off, and the bore diameter increases with a high rate. With the projectile volume increasing from 0 to 24%, the bore diameter of the muzzle increases from 12.67 to 12.81 mm. This increasing rate is much faster than that in the middle part. The muzzle is the final place where the linear and angular velocities of the projectile reach their highest

value, thus there is severe friction between the warheads and the bore surface. The Cr layer on the land lines in the muzzle was completely worn off, which results in a low riffling retention of $X = 1.2$ and weakens the guiding effect on projectiles.

The damage characteristics of the barrel are summarized as follows: (i) the main damage mode of the Cr layer from the tail to the muzzle varies from peeling to wearing; and (ii) the bore diameter in both the tail and muzzle increases with a high rate, while the damage rate in the middle part is comparatively low.

*4.2. Softening of Steel Matrix Near Bore Surface*

From the gun tail to the muzzle, the hardness increases gradually. The reasons for the softening of the tail matrix are as follows: (1) during the processing of barrel and after Cr plating, a stress–relief–annealing is generally needed for the gun tail, with temperatures between 500 and 600 °C and a soaking time of 1–3 h; (2) the gun tail suffers from the highest temperature in service. Feng et al. [24] calculated the bore surface temperature for a similar gun barrel (12.7 mm machine gun). Their calculation shows that after 120 shots, the bore surface temperature reaches 827 °C and then attenuates to 594 °C in 10 ms. Shooting temperature is also an important reason for the softening of the matrix.

From the bore surface to the outer surface of the barrel, the hardness increases gradually, which on the other hand, confirms that the shooting temperature causes the matrix to soften. According to the high bore surface temperature and pressure, it can be inferred that the bore surface was subjected to plastic deformation. Li et al. [25] declared that the bore surface of a rapid-fire artillery barrel suffers from plastic deformation along the radial direction within a thickness of about 0.467 mm. Under the action of high thermo-compression coupling stress, the bore surface softens. This increases the tendency of the plastic deformation of the matrix near the bore surface and increases the bore diameter, which will lead to a reduction in shooting accuracy.

## 5. Conclusions

In this study, the damage characteristics of a machine gun barrel were systematically investigated. The main results are as follows:

1.  The bore surface of the barrel presents different damage characteristics along the axial direction. In the gun tail, the Cr layer falls off completely, leaving the exposed matrix full of crack networks. The steel matrix without the protection of a Cr layer exhibits a higher damage rate. In the middle part, the Cr layer is well retained, with only a small part peeling off. This part of the barrel has a relatively small damage rate. In the muzzle, the Cr layer on the land lines is completely worn off, leaving the matrix exposed. The muzzle also exhibits a high damage rate.
2.  The matrix close to the bore surface is softened due to the high temperature and pressure caused by shooting. The hardness in the tail part is lower than those in both the middle and muzzle parts because the tail is subjected to higher temperatures during both stress–relief–annealing and firing processes. The softening of the matrix increases the tendency of the plastic deformation of the bore surface and increases the bore diameter, which will lead to a reduction in shooting accuracy.

**Author Contributions:** Z.-m.W.: formal analysis, writing—original draft; C.-d.H.: writing—review and editing, supervision, methodology; Y.-x.W.: investigation; H.-c.L.: review & editing, supervision; J.-s.L.: methodology; H.D.: methodology. All authors have read and agreed to the published version of the manuscript.

**Funding:** This research was funded by the Bottleneck Project of General Armament Department (grant number 30407).

**Informed Consent Statement:** Informed consent was obtained from all subjects involved in the study.

**Data Availability Statement:** Not applicable.

**Conflicts of Interest:** The authors declare no conflict of interest.

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
