# Peer review of "Failure Mechanism of Gun Barrel Caused by Peeling of Cr Layer and Softening of Bore Matrix"

_metals, doi:10.3390/met11020348_

Round 1
Reviewer 1 Report
The revised version of the manuscript is significantly better than the first version. All major comments have been taken into account. The disadvantages have been eliminated. In its current form, this manuscript can be accepted for publication.
Author Response
Thank you very much for your consideration and recognition.
Reviewer 2 Report
The authors edited their paper of a gun material.
In the following discussion, I am going to use the comment numbers introduced by authors for second referee. I also added more comments.
Comment 2: Readers expect to see the chemistry of the materials studied.
Comment on V1: The authors provided the chemistry without error bars. Please provide error bars.
Comment 3: Authors used both past tense and present tense in the experimental methods. I don’t know why. If not particular reason, I suggest use the past tense.
Authors replied “We have carefully and thoroughly proofread the manuscript to correct all the grammar.”
Comment on V1: I still see the mixed tense in the experimental methods. .
Comment 4: Authors should clarify who is the inventor and the producer of so-called “3D intelligent hyperfield microscopy”
Comment on V1: I am happy to see that the authors provide the vendor and exact name of their 3D intelligent hyperfield microscopy”. This microscope was introduced before 2015. Therefore, detail description of the microscope is not needed in the article.
Comment 5: Authors stated “The long crack spacing was uniformly distributed to the naked eye, and the Cr layer was completely peeled ” From the picture, I cannot see the evidence of Cr layer peeling.
Comment on V1: I am afraid that it is still not clear to me.
Comment 5: Authors continue to comment “The damage of Cr layer on land of MG4 is more serious than that on groove, the thickness of Cr layer on land is thinner than that of MG3 sample, the thickest Cr layer is about 130 μm. ” The image has a resolution worse than 1 μm. I am not sure how authors achieve the measurement of 130 μm accuracy.
Comment on V1: I would like to see the error bar for the measurement.
Comment 9: Authors stated in conclusion “ A newly equipped named 3DIHM ” It appears that authors thought this was the new in their paper. If so, the authors should clarify who invented and produced this microscope.
Comment on V1: I am glad to see that authors removed “newly equipped “ in the new manuscript. But what is new in the current manuscript?
Comment 10: Authors stated that “ The cracks in Cr layer expand continuously to the matrix, causing the matrix damage. Cracking is an important way of stress release and an important cause of bore diameter increase. ” in conclusion and “Cracking is an important way of stress release and an important cause of bore diameter increase.” in abstract. Generally speaking, authors repeated what they said in abstract and conclusion word by word. This is similar to self plagiarism and should be corrected.
How did readers know it is a stress release mechanism if the authors didn’t mention this at all in the manuscript? Authors should provide more evidence to proof this.
Comment on V1: It appears that the authors didn’t answer my second portion of the question.
Comment 11It appears to me that the cyclic hoop-tensile- stress may have caused the low cycle fatigue cracks. I recommend authors to refer the following papers in order to understand the cause of the cracks and crack propagations.
doi.org/10.3390/met9070722
doi.org/10.1063/1.3402339
10.1109/TASC.2009.2039556
10.1109/TASC.2013.2289892
Comment on V1. It appears that this portion of comment was missed by the authors.
Comment 12:
In the introduction, the authors stated “The mechanism of bore damage was analyzed from a new perspective.” Please clarify what is a new perspective.
Comment 13: Author stated “the exposed matrix is full of turtle cracks”. This is difficult to understand.
Comment 14: Authors stated “In the mouth, the Cr layer on land is worn and the steel matrix is exposed; the mouth has a high damage rate. “ English needs a lot of improvement in order for readers to understand.
”
Comment 15: Authors stated “…both qualitative tempering and firing…. ” I am afraid that I don’t understand.
Generally speaking, I recommend the authors to discover more science from their work for next submission. Current style is similar to a technical report.
Reviewer 3 Report
acceptable as-revised
Author Response
Thank you very much for your consideration and recognition
Round 2
Reviewer 2 Report
Comment 5: Authors stated “The long crack spacing was uniformly distributed to the naked eye, and the Cr layer was completely peeled ” From the picture, I cannot see the evidence of Cr layer peeling.
Comment on V1: I am afraid that it is still not clear to me.
Response 4: Before shooting, the whole bore surface is covered with a Cr layer. Figure 3(a) shows the cross-sectional morphology of sample MG1 (tail region) after shooting. In the picture, no Cr layer is observed. Thus, we can conclude that the Cr layer has been peeled off during shooting. Modifications to the description of Figure 3(a) have been made in the revised manuscript.
Comment on V2: The enlarged images help. Can authors enlarge it further?
Comment 12:
In the introduction, the authors stated “The mechanism of bore damage was analyzed from a new perspective.” Please clarify what is a new perspective.
Comment 13: Author stated “the exposed matrix is full of turtle cracks”. This is difficult to understand.
Reply “Before shooting, the surface of the bore is covered with a Cr layer. After shooting, the Cr layer in the tail peels off, leaving the matrix exposed. Fig.4 (a) and Fig. 5 (a) in the manuscript are images of exposed matrix. Turtle cracks are observed to spread all over the region in these figures.”
Comment on V2: I don’t think turtle crack is a proper English. Authors can say surface cracks, or crack networks or similar. Overall the English still needs a lot of work. Please find a professional English editor or English speaking person to do further editing before next submission. This applies to my other comments.
Comment 14: Authors stated “In the mouth, the Cr layer on land is worn and the steel matrix is exposed; the mouth has a high damage rate. “ English needs a lot of improvement in order for readers to understand.”
Comment 15: Authors stated “…both qualitative tempering and firing…. ” I am afraid that I don’t understand.
Generally speaking, I recommend the authors to discover more science from their work for next submission. Current style is similar to a technical report.
Author Response
Please see the attachment.

This manuscript is a resubmission of an earlier submission. The following is a list of the peer review reports and author responses from that submission.
Round 1
Reviewer 1 Report
This paper deals with failure analysis of an artificially degraded machine gun component. Although failure mechanism study is one of the important fields of metal research for quality control of the product and reliable application, this paper is not suitable for publication in this journal due to the following critical reasons.
Experimental data and their analysis are not enough to reveal detailed failure mechanism of the component investigated. The use of 3D hyperfield microscopy does not always mean that better scientific information for metal study can be offered, although it is a convenient tool for easy observation of macro area. The followings can be considered for further investigation; detailed microscopic analysis of the damaged layer, detailed compositional investigation, direct temperature profiling, observation and analysis for detailed wear mechanism (i.e., what kind of wear?), details on the Cr layer. The hardness difference between the positions investigated seems not to be high to induce severe damage. Detailed explanations are required for the low surface hardness value. Little original discussion point is also found in the paper. Finally, the conclusions are not well supported by the results presented. Case report-related publication may be recommended for this manuscript, rather than material research.
Reviewer 2 Report
The authors described their studies of a gun material.
The authors stated “The sample GM2 studied in this paper is a longitudinal section sample, and the rest are ring samples.” It GM should be MG?
Readers expect to see the chemistry of the materials studied.
Authors used both past tense and present tense in the experimental methods. I don’t know why. If not particular reason, I suggest use the past tense.
Authors should clarify who is the inventor and the producer of so-called “3D intelligent hyperfield microscopy”
Authors stated “The long crack spacing was uniformly distributed to the naked eye, and the Cr layer was completely peeled ” From the picture, I cannot see the evidence of Cr layer peeling.
Authors continue to comment “The damage of Cr layer on land of MG4 is more serious than that on groove, the thickness of Cr layer on land is thinner than that of MG3 sample, the thickest Cr layer is about 130 μm. ” The image has a resolution worse than 1 μm. I am not sure how authors achieve the measurement of 130 μm accuracy.
Fig 3, I think “idiameter” should be “diameter”?
In Fig. 10. Radial hardness of barrel tail, middle and muzzle, can authors include error bars in the figure?
Authors stated in conclusion “ A newly equipped named 3DIHM ” It appears that authors thought this was the new in their paper. If so, the authors should clarify who invented and produced this microscope.
Authors stated that “ The cracks in Cr layer expand continuously to the matrix, causing the matrix damage. Cracking is an important way of stress release and an important cause of bore diameter increase. ” in conclusion and “Cracking is an important way of stress release and an important cause of bore diameter increase.” in abstract. Generally speaking, authors repeated what they said in abstract and conclusion word by word. This is similar to self plagiarism and should be corrected.
How did readers know it is a stress release mechanism if the authors didn’t mention this at all in the manuscript? Authors should provide more evidence to proof this.
It appears to me that the cyclic hoop-tensile- stress may have caused the low cycle fatigue cracks. I recommend authors to refer the following papers in order to understand the cause of the cracks and crack propagations.
Enhancement of fatigue endurance by Al-Si coating in hot-stamping boron steel sheet
Y Li, et al
Metals 9 (7), 722 (2019)
FATIGUE PROPERTIES OF MODIFIED 316LN STAINLESS STEEL AT 4 K FOR HIGH FIELD CABLE‐IN‐CONDUIT APPLICATIONS
VJ Toplosky, et al
AIP Conference Proceedings 1219 (1), 9-16 (2010)
Fatigue property examinations of conductors for pulsed magnets
K Han, et al
IEEE Transactions on Applied Superconductivity 20 (3), 1463-1466 (2010)
Fatigue properties of Cu–Nb conductor used for pulsed magnets at the WHMFC, QQ Sun, …, et al, IEEE Transactions on Applied Superconductivity ( Volume: 24, Issue: 3, June 2014)
Article Sequence Number: 0501404
DOI: 10.1109/TASC.2013.2289892.
Reviewer 3 Report
The article is certainly interesting from a practical point of view and relevant for science. However, the presented results require significant improvement.
Comments:
- The literature review can be expanded to reveal more clearly the work done to date in this area.
- A typo in Figure 3. "Inner idameter".
- Figure 6 is of very poor quality. In addition, unreadable designations along the vertical and horizontal axes make it difficult to adequately analyze the given result.
- Figure 9. Poor quality. It could be completely removed. Moreover, the data for MG1 and MG7 are partially duplicated due to SEM images. Deformation is still not distinguishable on such metallography ...
- Graph in Figure 10. Measurement of hardness from the surface at a distance of 0 and immediately ~ 1 mm is too rough. A more detailed measurement should be made near the surface of the barrel. This can be done by shifting the indenter penetration region in a checkerboard pattern. Or by indentation not along four lines, but along 8-16 lines with the displacement of the first imprint from the surface by 50 μm, 150 μm, etc.
- The authors refer a lot to the new 3DIHM method. However, the results themselves given using this method are limited to one image of poor quality. All reasoning is based mainly on the analysis of SEM images, EMF and conventional metallography with hardness measurement.
The authors should cite more results obtained using this method and conduct a deeper analysis of these data. Worn / eroded topography is a very interesting area of research.
- Author Contributions: the role of the authors should be indicated, not the fact that they have read the article. The MDPI website has clear requirements for the design of this item.